# Targeting Lymphedema in Overweight Breast Cancer Survivors: A Pilot Randomized Controlled Trial of Diet and Exercise Intervention

**DOI:** 10.3390/nu17172768

**Published:** 2025-08-27

**Authors:** Yolanda Ruiz-Molina, Marina Padial, María del Mar Martín-Bravo, María García-Olivares, Nuria Porras, Alejandro Chicharro, Javier Mora-Robles, Andrés González-Jiménez, Corina Verónica Sasso, Gabriel Olveira

**Affiliations:** 1Servicio de Medicina Física y Rehabilitación, Hospital Virgen de la Victoria, 29010 Malaga, Spain; yorumoce@hotmail.com; 2Departamento de Medicina y Dermatología, Universidad de Málaga, 29010 Malaga, Spainverosasso86@gmail.com (C.V.S.); 3Servicio de Endocrinología y Nutrición, Instituto de Investigación Biomédica de Málaga y Plataforma en Nanomedicina–IBIMA Plataforma Bionand, Hospital Regional Universitario de Málaga, 29010 Malaga, Spain; 4Servicio de Medicina Física y Rehabilitación, Instituto de Investigación Biomédica de Málaga y Plataforma en Nanomedicina–IBIMA Plataforma Bionand, Hospital Regional Universitario de Málaga, 29010 Malaga, Spain; 5Servicio de Cardiología, Instituto de Investigación Biomédica de Málaga y Plataforma en Nanomedicina–IBIMA Plataforma Bionand, Hospital Regional Universitario de Málaga, 29010 Malaga, Spain; 6Plataforma de Bioinformática, Instituto de Investigación Biomédica de Málaga y Plataforma en Nanomedicina–IBIMA Plataforma Bionand, 29590 Malaga, Spain; bioinformatica@ibima.eu; 7Centro de Investigación Biomédica en Red (CIBER) de Diabetes y Enfermedades Metabólicas Asociadas, Instituto de Salud Carlos III, 29071 Malaga, Spain

**Keywords:** breast cancer-related lymphedema, overweight, lifestyle intervention, weight loss, Mediterranean Diet, supervised exercise, morphofunctional status

## Abstract

**Background/Objectives**: Breast cancer-related lymphedema (BCRL) is more prevalent and severe in women with overweight or obesity. This study evaluated the effect of a comprehensive lifestyle intervention—comprising supervised exercise, a hypocaloric Mediterranean diet, and optional meal replacement—on lymphedema outcomes in this population. **Methods**: In this pilot randomized controlled trial, 112 women with BCRL and BMI 25–40 kg/m^2^ were assigned to an intervention group—receiving supervised resistance and aerobic training, dietary counseling, and optional high-protein meal replacement—or to a control group with standard advice. The primary outcome was change in affected limb volume at 3 and 6 months. Secondary outcomes included morphofunctional parameters, muscle strength, dietary intake, and serum levels of cytokines (IL-1β, IL-6, IL-10, and TNF-α). Analyses also explored outcomes according to whether participants achieved ≥ 5% weight loss. **Results**: Ninety-four participants completed the trial (intervention n = 43, control n = 51). At 6 months, women who achieved ≥5% weight loss had greater reductions in affected limb volume (−664.9 ± 362.1 mL vs. −395.6 ± 596.9 mL). The intervention group showed significantly greater improvements in BMI (−1.14 ± 1.22 kg/m^2^), waist circumference (−3.59 ± 4.6 cm), triceps skinfold (−4.61 ± 3.02 mm), fat mass (−2.38 ± 2.75 kg), extracellular water (−0.58 ± 0.85 L), and quadriceps strength (+7.1 ± 9.7 kg). No significant changes were observed in circulating cytokines. **Conclusions**: In this pilot randomized controlled trial, a structured dietary and exercise intervention improved morphofunctional outcomes in overweight women with BCRL. Weight loss of ≥5% emerged as a potentially relevant therapeutic target that may inform the design of future studies aimed at optimizing lymphedema management.

## 1. Introduction

Secondary lymphedema due to breast cancer (BCRL) is a common chronic complication following oncological treatment, particularly in women undergoing axillary lymphadenectomy [1,2]. It is characterized by the accumulation of protein-rich interstitial fluid, inflammation, and fibrosis, which affect mobility and quality of life (QoL) [3,4,5]. Its prevalence typically ranges from 20% to 24% two years post-surgery [6,7]. However, this prevalence significantly increases to nearly 40% in women with overweight or obesity [8,9]. Obesity is not only a risk factor for the development of BCRL but also exacerbates its progression due to chronic inflammation and increased lymphatic system overload [10,11,12].

Various therapeutic strategies, such as lymphatic drainage, physiotherapy, and compression, have been evaluated for BCRL management, but their effectiveness in women with obesity is limited [13,14]. Given this limitation, researchers have explored the potential benefits of weight loss and physical activity. Evidence regarding the impact of weight loss on BCRL is mixed; while some studies have reported significant reductions in lymphedema volume associated with weight loss [15,16], others have found limited effects on interlimb volume differences [10] or BCRL risk [16,17]. Nevertheless, a 5% reduction in body weight has been shown to confer meaningful health benefits, including improvements in cardiovascular risk factors, morbidity, and mortality, and is considered a clinically relevant target in individuals with overweight and obesity [18]. In fact, recent studies suggest that such a weight reduction may also contribute to improved lymphedema outcomes, including limb volume reduction and symptom relief in women with BCRL [19].

Complementing this findings, structured physical activity has proven effective in enhancing QoL and reducing inflammation [20,21]. In this context, the WISER Survivor Trial [22] investigated the effects of home-based exercise and weight loss interventions in women with post-breast cancer lymphedema. The study found that these interventions did not significantly improve BCRL outcomes. However, the findings suggest that supervised, facility-based exercise programs may be more beneficial, warranting further investigation into structured exercise strategies for managing BCRL, particularly in women with obesity. Furthermore, adherence to a hypocaloric Mediterranean Diet (MedDiet) has been associated with reductions in inflammation and cardiovascular risk factors [23,24,25], and the use of meal replacement products has proven effective in supporting weight loss and dietary adherence [26,27].

Building on this evidence, the present study aimed to evaluate the effect of a comprehensive lifestyle intervention—consisting of a supervised exercise program, a hypocaloric MedDiet, and optional meal replacement supplementation—on lymphedema outcomes in women with BCRL who are overweight or obese. We hypothesized that the intervention would result in significant weight loss and reductions in lymphedema volume.

## 2. Materials and Methods

### 2.1. Study Design and Participants

This is a prospective, randomized, open-label pilot clinical trial with two parallel groups conducted at the Regional Hospital of Málaga (HRM) between June 2021 and June 2023 This RCT was registered on ClinicalTrials.gov (NCT04974268) on 23 June 2021.

The inclusion criteria required women aged 18 to 79 with a diagnosis of BCRL, defined by a volume increase of more than 200 mL in the affected limb compared to the contralateral limb (calculated using the truncated cone formula [28]). Eligible participants must have been referred to rehabilitation at least six months after completing chemotherapy and radiotherapy, with no manual lymphatic drainage received in the six months prior to the intervention. Additionally, a Body Mass Index (BMI) between 25 and 40 kg/m^2^ was required. Exclusion criteria included traumatic, neurological, rheumatological, or cardiovascular conditions that prevent training; structured lymphedema in Phase IIIB; metastatic disease; severe mental disorders; severe heart disease (including aortic stenosis, hypertrophic cardiomyopathy, or a left ventricular ejection fraction < 35%); voluntary or involuntary weight loss of more than 10% in the last three months; illiteracy; alcohol abuse or other substance dependencies; and pregnancy.

The participants provided written informed consent, and the study protocols were approved by the Provincial Research Ethics Committee of Málaga (protocol code “EJERDIETLINF”) on 26 February 2020, in full compliance with the principles of the Declaration of Helsinki.

### 2.2. Randomization and Intervention

Patients were randomly assigned to either the intervention or control group in a 1:1 ratio, using a computer-generated random number table:Intervention group: participants received a 12-week intensive weight loss program, which included: supervised exercise (2 sessions per week, 90 min each) with resistance and aerobic training; dietary intervention based on a hypocaloric MedDiet, with optional daily meal replacement (Bi1 Bificare^®^, Adventia Pharma S.L., Las Palmas de Gran Canaria, Spain); and manual lymphatic drainage and compression garment use for patients with excess limb volume > 600 mL.Control group: participants received standard recommendations, which included: unsupervised aerobic exercise (150 min per week); standard MedDiet (1600–1800 kcal); and manual lymphatic drainage and compression garment use for patients with excess limb volume > 600 mL.

Before starting, participants in the intervention group underwent medical screening, including cardiopulmonary exercise testing and echocardiography (Teichholz method) to assess exercise capacity and ventricular function.

The supervised exercise program combined:Warm-up (10 min)Strength training (30 min) targeting upper and lower limbs, progressing from 2 to 3 sets of 10 reps per exerciseAerobic exercise (20 min) at moderate-to-high intensityCool-down (5 min)Stretching (15 min)

Participants were encouraged to perform additional aerobic exercise at home on non-training days. Exercise intensity was based on cardiopulmonary test results (75–80% of peak watts) and resistance training at >80% of 1RM, adjusted weekly.

The dietary program aimed for a 10% reduction in body weight over six months, with a minimum target of 5%. It included:Seven individual sessions and three group sessionsEducation on a hypocaloric MedDiet, including meal plans, recipes, and strategies for managing cravings and stress.Optional one daily meal replacement (Bi1 Bificare^®^).

The Bi1 Bificare^®^ supplement is a normocaloric, high-protein complete oral nutritional supplement formulated for clinical use. Its fat source is extra virgin olive oil, and it contains added leucine and omega-3 fatty acids (EPA and DHA) to support muscle maintenance and anti-inflammatory processes. It is fortified with 3.3 billion units of the probiotic strain *Bifidobacterium* animalis subsp. lactis CECT 8145 (BPL1). This strain—whether alive or heat-inactivated—has been associated with improvements in gut microbiota and reductions in cardiometabolic risk factors such as abdominal fat, glucose metabolism alterations, and hypertension [29]. The nutritional composition is detailed in Appendix A.

All patients were evaluated at baseline, 3 months, and 6 months for a range of clinical and lifestyle variables. The primary outcome was the change in lymphedema volume. Secondary outcomes included assessments of morphofunctional status, dietary intake and adherence to the Mediterranean Diet, biochemical and inflammatory markers, and physical activity.

### 2.3. Outcome Measures

#### 2.3.1. Affected Limb Volume and Circumference

Limb volume and circumference were assessed using a tool created by the Spanish Society of Rehabilitation and Physical Medicine (SERMEF), the SERMEF calculator (Sociedad Española de Rehabilitación y Medicina Física [SERMEF], Madrid, Spain) [30], which applies the truncated cone formula to estimate limb volume accurately.

#### 2.3.2. Morphofunctional Assessment

BMI was calculated as weight in kilograms divided by height squared (in meters). Mid-arm circumference in non-affected arm was measured using a non-stretchable tape measure, and triceps skinfold thickness was assessed with a constant pressure skinfold caliper (Holtain Limited, Crosswell, Crymych, Pembrokeshire, Wales, UK) by the same researcher. Each measurement was taken three times, and the values were averaged.

Bioelectrical impedance analysis (BIA) was conducted using the TANITA MC980MA analyzer (Tanita Corporation, Tokyo, Japan). Measurements were performed with participants in a standing position, following standardized procedures. The device’s integrated software was used to assess weight, fat mass (FM), fat-free mass (FFM), muscle mass (MM), extracellular water (ECW), and phase angle (PhA). In addition, the fat-free mass index (FFMI) was calculated as FFM divided by height squared (kg/m^2^). Segmental analysis was used to obtain FM, FFM, and skeletal muscle mass (SMM) specifically in the affected and unaffected arms.

Muscle strength was measured using a Jamar Handgrip dynamometer (Jamar Handgrip; Asimow Engineering Co., Los Angeles, CA, USA) on the non-affected arm. Each measurement was performed three times, recording both the average and maximum values. Reference values for handgrip strength in the Spanish population were applied [31]. Additionally, quadriceps isometric strength (QIS) was assessed using a dynamometer Commander PowerTrack II dynamometer (Commander PowerTrack II; JTECH Medical, Midvale, UT, USA).

#### 2.3.3. Dietary Intake and Adherence to MedDiet

Dietary assessment was conducted using a 4-day prospective dietary record, including three weekdays and one weekend day. Participants were instructed to record all foods and beverages consumed, with portion sizes estimated using household measures. The data were analyzed using Dietstat^®^ (FIMABIS, Málaga, Spain; institutional version 2016), a software application developed by our research group for nutritional evaluation [32]. This tool was used to calculate energy and nutrient intake, as well as adherence to dietary recommendations.

Adherence to the Mediterranean Diet was assessed using the 17-item PREDIMED-Plus questionnaire [33], a modified and energy-restricted version of the original validated PREDIMED score [34]. This tool was specifically adapted to reflect compliance with a hypocaloric Mediterranean dietary pattern, intended for weight loss interventions. Each item aligned with recommended dietary behaviors (e.g., preference for olive oil, high intake of fruits, vegetables, legumes, fish, and whole grains; low intake of red meat, sugary drinks, and processed foods) scored 1 point if fulfilled, and 0 otherwise, yielding a total score ranging from 0 to 17. Higher scores indicated greater adherence to the dietary protocol.

Adherence to the supervised exercise program was assessed prospectively using a predefined four-level ordinal scale (poor, fair, good, excellent) based on attendance to scheduled sessions and completion of the prescribed sets/repetitions, recorded at each follow-up visit.

#### 2.3.4. Blood Parameters

Blood samples were collected after an overnight fast at baseline and after three and six months. One aliquot was analyzed immediately in a hospital-based autoanalyzer to measure hemoglobin, hematocrit, albumin, creatinine, urea, lipid profile (total cholesterol, HDL, LDL, triglycerides), iron metabolism markers (iron, ferritin, transferrin), electrolytes (sodium, potassium, chloride, calcium), and glucose metabolism markers (insulin and HbA1c). Vitamin D and albumin-corrected calcium levels were also assessed.

Cytokine concentrations were quantified using a multiplex bead-based immunoassay (Human ProcartaPlex™ Mix & Match 4-plex, Thermo Fisher Scientific, Waltham, MA, USA), intra-assay CV < 10%; inter-assay CV < 15%; according to the manufacturer’s instructions. The following cytokines were measured: interleukin-1 beta (IL-1β), interleukin-6 (IL-6), interleukin-10 (IL-10), and tumor necrosis factor alpha (TNF-α). Measurements were performed in duplicate using 25 microliters of human serum per well.

### 2.4. Sample Size and Power Calculation

According to the study by Schmitz et al. [22], an average volume reduction of 274 mL was observed in the affected arm with a standard deviation of 407 mL in the group receiving diet and exercise. Assuming a 95% confidence interval and 80% power to detect a volume reduction of at least 274 mL, it was estimated that 36 patients per group (72 total) would be required. Taking into account a 20% loss to follow-up, it was necessary to recruit at least 44 patients per group (88 total).

### 2.5. Statistical Analysis

The study population was initially divided into two distinct groups: intervention and control. Baseline characteristics were compared between groups using the Student’s t-test when the assumption of normality was satisfied. In cases where normality was not met, the Mann–Whitney U test was employed. For qualitative variables, comparisons were conducted using a chi-square test approximation.

Efficacy analyses were performed following both intention-to-treat (ITT) and per-protocol (PP) approaches. The PP population included participants who adhered to the allocated intervention and completed the study assessments, and these results are presented as the main analyses in the manuscript. In parallel, ITT analyses were conducted including all randomized participants with available outcome data, regardless of adherence; these results are provided in the Appendix A.

For the analysis of lymphedema volume, participants were further categorized in a post hoc analysis based on the degree of weight loss achieved during the study. A reduction of ≥5% from initial body weight was considered clinically meaningful. Accordingly, participants were classified into two groups: weight loss group (WL group) and non-weight loss group (Non-WL group). In contrast, morphofunctional parameters were analyzed using the original intervention and control groups. To assess the interaction between different time points and study groups, a longitudinal analysis was conducted to evaluate changes in these parameters across three assessment points. Between-group comparisons at each time point were carried out using either the Student’s *t*-test or the Mann–Whitney U test, depending on whether the data met the assumptions of normality.

Changes in dietary intake over time were assessed using a one-way ANOVA when the assumption of normality was met. If significant differences were identified, pairwise comparisons were performed using Tukey’s Honest Significant Difference (HSD) test. When the assumption of normality was violated, the Kruskal–Wallis test was applied, followed by post hoc comparisons using the Mann–Whitney U test.

## 3. Results

From 130 patients referred to the clinic, 112 were included in the study, with 94 completing the 6-month follow-up (intervention *n* = 43, control *n* = 51). The study flowchart is presented in Figure 1. In the intervention group, most withdrawals occurred prior to the 3-month follow-up and were primarily due to difficulties attending the supervised exercise sessions at the hospital, which posed logistical barriers for some participants.

At baseline, no significant differences were observed between the intervention and control groups in terms of age, marital status, education level, cancer stage, type of surgery, radiotherapy, or comorbidities. This indicates that both groups were comparable at the start of the study (Table 1).

Main results are presented for the PP population. Complementary ITT analyses are available in the Appendix A. Analyses stratified by achievement of ≥5% weight loss, irrespective of randomization group, are presented as post hoc analyses.

### 3.1. Limb Circumference and Volume

At baseline, both groups (WL/non-WL) presented similar limb volume and circumference values (Table 2). Over time, participants in the WL group (*n* = 27; 19 from the intervention group and 8 from the control group) showed significantly greater reductions in affected limb volume at both 3 (−380.89 vs. −153.55) and 6 months (−664.91 vs. −395.58) compared to non-WL group, with a progressive pattern and larger decreases observed at 6 months.

Significant between-group differences were also found in several circumferential measures of the affected limb. Reductions were observed at the 65% level at 3 months (−1.24 vs. −0.27) and 6 months (−2.2 vs. −0.96), mid-arm at 3 months (−1.26 vs. −0.38) and 6 months (−2.04 vs. −1.06), and forearm at 3 months (−0.70 vs. 0.04) and 6 months (−1.3 vs. −0.36). At the wrist, a significant reduction was observed only at 3 months (−0.54 vs. −0.28), with no further change at 6 months.

However, a significant between-group difference was observed in healthy limb volume at 6 months (−484.91 vs. −266.38), with greater reductions in the WL group. Similarly, circumference reductions at the 65% and mid-arm levels of the healthy limb were significantly greater in WL group at both 3 and 6 months, as well as at the elbow. No significant between-group differences were found at the forearm or wrist levels of the healthy arm.

### 3.2. Morphofunctional Assessment

At baseline, no significant differences were observed between groups in morphofunctional parameters (Table 3). Over time, the intervention group showed significantly greater reductions in BMI at 3 months (−1.17 vs. −0.54) and 6 months (−1.14 vs. −0.53), with additional reductions from 3 to 6 months.

Triceps skinfold decreased more in the intervention group at 3 months (−2.9 vs. −1.12) and 6 months (−4.61 vs. −1.39), with further reductions between timepoints. Waist circumference decreased more in the intervention group at 3 months (−2.93 vs. −1.16, *p* < 0.05) and 6 months (−3.59 vs. −2.19), with significant progression between timepoints.

Regarding body composition measured by BIA, FM decreased more in the intervention group at 3 months (−1.84 vs. −0.81) and 6 months (−2.38 vs. −1.05), with further reduction between timepoints. ECW also decreased more at 6 months (−0.58 vs. −0.24), with significant progression. FFMI remained stable between groups but decreased within the intervention group from 3 to 6 months (−0.31 vs. −0.10). In the affected arm, small but significant decreases in FM were observed from 3 to 6 months in both groups (−0.08 vs. −0.09), with no significant between-group differences in FFM or SMM.

Muscle function, assessed by dynamometry, showed greater improvement in QIS in the intervention group (7.10 vs. −3.35) at 6 months, while HGS increased modestly in both groups without significant differences.

In the intervention arm, adherence to supervised exercise was assessed in 28 participants (15 missing assessments). Overall, 22/28 (78.6%) showed high adherence (Good 15 [53.6%]/Excellent 7 [25.0%]), while 6/28 (21.4%) showed low adherence (Poor 4 [14.3%]/Fair 2 [7.1%]).

### 3.3. Dietary Composition

At baseline, dietary intake was similar between groups (Table 4). Over time, the intervention group showed greater reductions in energy intake at 3 months (−290.22 vs. −43.35) and 6 months (−216.22 vs. −19.03), with further reductions between timepoints.

Total fat intake decreased in the intervention group at 3 months (−14.88 vs. +2.94) and 6 months (−9.66 vs. 5.32), while it increased in the control group. This was particularly evident for saturated fat intake (−4.36 vs. 1.61), and was also reflected in cholesterol intake, which decreased in the intervention group at 6 months (−50.62 vs. 28.53), while it increased in the control group. Similarly, omega-3 intake increased in the intervention group at 3 months (0.92 vs. 0.20) and 6 months (0.64 vs. 0.37), while omega-6 intake decreased at 3 months (−2.15 vs. 0.76) and at 6 months (−1.66 vs. 2.49).

Glycemic load also decreased more in the intervention group at 3 months (−25.93 vs. −6.36) and 6 months (−19.67 vs. −3.74). Glycemic index was lower in the intervention group at both 3 months (−12.1 vs. 0.31) and 6 months (−6.87 vs. 0.39). Similarly, fiber intake increased more in the intervention group at 3 months (6.43 vs. 1.60), although the difference narrowed by 6 months (5.34 vs. 3.27).

No significant differences were observed in protein, carbohydrates, EPA, DHA, or calcium intake between groups.

Notably, of the 43 participants in the intervention group, 32 used the meal replacement supplement at least once during the study, and 19 reported consuming at least half of the recommended doses (≥90 units) over the 6-month period.

### 3.4. Biochemical Markers

No significant between-group differences were observed over time in biochemical parameters (Appendix A), including hemoglobin, hematocrit, albumin, creatinine, urea, lipid profile (total cholesterol, HDL, LDL, triglycerides), or markers of iron status (iron, ferritin, transferrin). Electrolytes such as sodium, potassium, chloride, and calcium also remained stable throughout the intervention in both groups, with no clinically relevant changes. Similarly, no significant changes were observed in insulin levels or HbA1c, although a slight increase in was noted in the control group at 3 months (0.40 vs. 0.07), which did not persist over time.

Additionally, circulating levels of inflammatory cytokines—IL−1, IL-6, IL-10, and TNF-α—were measured. However, no significant differences were found between groups or across timepoints, suggesting that the intervention did not produce a measurable systemic inflammatory response within the timeframe of the study. Detailed cytokine data are presented in Appendix A.

## 4. Discussion

In this pilot clinical trial, women with BCRL and overweight who achieved a weight loss of more than 5% of their initial body weight experienced greater improvements in lymphedema-related outcomes, including significant reductions in limb volume and circumference. Morphofunctional improvements were observed in the intervention group, including higher reductions in BMI, waist circumference, skinfold thickness, FM, and ECW, as well as improved QIS. These changes were accompanied by a healthier dietary profile, with lower intake of energy, saturated fat, glycemic load, and cholesterol, and higher intake of fiber and omega-3 fatty acids. However, no significant differences were found between groups in biochemical or inflammatory markers over time.

Importantly, the consistency of results across both PP (main analyses) and ITT (Appendix A) populations reinforces the robustness of these findings.

To better reflect the influence of individual weight changes on clinical outcomes, participants were analyzed not only by study group (intervention vs. control), but also according to whether or not they achieved clinically relevant weigh loss. This post hoc classification was based on growing evidence suggesting that limb volume reduction in BCRL is more closely linked to weight loss than to the intervention strategy itself [22,35]. Although only the intervention group received a structured, supervised program, the control group also benefited from dietary and physical activity counseling, which enabled some participants to achieve meaningful weight loss. This likely contributed to the lack of clear differences in lymphedema outcomes when analyzed by group allocation alone. 

Furthermore, as observed in prior studies [10,15], weight loss in our cohort led to reductions in both affected and unaffected arm volumes. As a result, the interlimb volume difference—a commonly used marker of BCRL severity—did not significantly differ between groups. These findings underscore the limitations of using volume asymmetry alone to evaluate lymphedema changes, and support the use of absolute volume reduction, particularly in the affected limb, as a more sensitive and clinically relevant indicator of improvement. However, it is important to note that obesity is not the only factor influencing lymphedema. In advanced cases, persistent fibrosis, chronic inflammation, or irreversible lymphatic damage may limit clinical improvement despite weight loss, underscoring the need for a comprehensive, multimodal approach [36,37].

Moreover, the morphofunctional improvements observed in the intervention group are consistent with prior findings. A recent systematic review by Wang et al. [38] reported that lifestyle interventions combining dietary restriction and physical activity significantly reduce body weight, waist circumference, FM, and BMI in breast cancer survivors, without compromising lean mass. In our study, these improvements were accompanied by a healthier dietary profile, characterized by reduced intake of saturated fat, glycemic load, and cholesterol, alongside increased fiber and omega-3 fatty acid intake. Although systemic inflammatory markers remained unchanged, the reduction in ECW suggests that localized or early anti-inflammatory responses may occur prior to detectable systemic changes, or that a longer intervention period may be required to observe broader effects.

Lastly, our findings reinforce the safety and value of including resistance training in BCRL management. Contrary to earlier recommendations that discouraged upper-body exercise, accumulating evidence supports the role of supervised resistance training in improving functional capacity and reducing symptom burden without worsening lymphedema [39,40,41]. In our study, muscle strength—particularly in the quadriceps—increased significantly in the intervention group, without adverse effects on limb volume. These results further support the integration of structured strength training into comprehensive treatment strategies for women with BCRL.

This study has several strengths. It is one of the few randomized controlled trials to evaluate a comprehensive lifestyle intervention—including supervised exercise, a hypocaloric Mediterranean Diet, and optional meal replacement—in women with BCRL and overweight or obesity. The integration of objective measures of limb volume, morphofunctional status, dietary intake, and biochemical markers allowed for a thorough assessment of intervention effects. Additionally, the classification of participants according to weight loss status provided important insights into the role of weight dynamics in lymphedema improvement. Notably, participants performed the supervised exercise sessions without compression garments, and no increases in limb volume or episodes of infection were observed during the study. This supports the safety and feasibility of strength training without compression in selected patients with BCRL and highlights a practical advantage for implementation in real-world clinical settings.

However, some limitations should be acknowledged. The sample size was modest, and the study was not powered to detect small differences in all secondary outcomes, particularly inflammatory or biochemical markers. The follow-up period, although sufficient to observe changes in limb volume and body composition, may have been too short to detect meaningful systemic changes in inflammation. The open-label design and the potential for “intervention spillover” to the control group may have attenuated between-group differences. Furthermore, the voluntary use of meal replacements and variability in attendance at supervised sessions introduced heterogeneity in adherence. While participants were randomized, the post hoc classification by ≥5% weight loss was exploratory and may have introduced selection bias, limiting causal inference in those comparisons. Nevertheless, the parallel ITT analyses confirmed the same overall trends, supporting the validity of the conclusions. Additionally, weight loss alone may not fully resolve lymphedema in cases with chronic fibrosis, persistent inflammation, or irreversible lymphatic damage, which could have attenuated the observed effects in some participants. Taken together, these factors indicate that this work should be interpreted as a preliminary, pilot randomized controlled trial, and the findings should be confirmed in larger, longer-term studies.

## 5. Conclusions

This pilot study demonstrates that a structured dietary and exercise intervention improved morphofunctional outcomes in women with BCRL. Weight loss of ≥5% was associated with significant reductions in the volume of the affected limb, although reductions were also observed in the healthy arm, underscoring the systemic benefits of weight loss on fluid balance. In addition to these lymphedema-specific improvements, the intervention led to favorable changes in body composition and dietary quality. Notably, even in the absence of structured, professionally supervised programs, basic counseling on diet and physical activity—as provided to the control group—was associated with clinically relevant benefits in some participants. These preliminary findings indicate that achieving a ≥5% weight loss may represent a relevant therapeutic target and provide a foundation for future studies aimed at optimizing lymphedema management. While guided multidisciplinary strategies are ideal, simple and accessible lifestyle advice may also contribute meaningfully to the management of BCRL when resources are limited.

## Figures and Tables

**Figure 1 nutrients-17-02768-f001:**
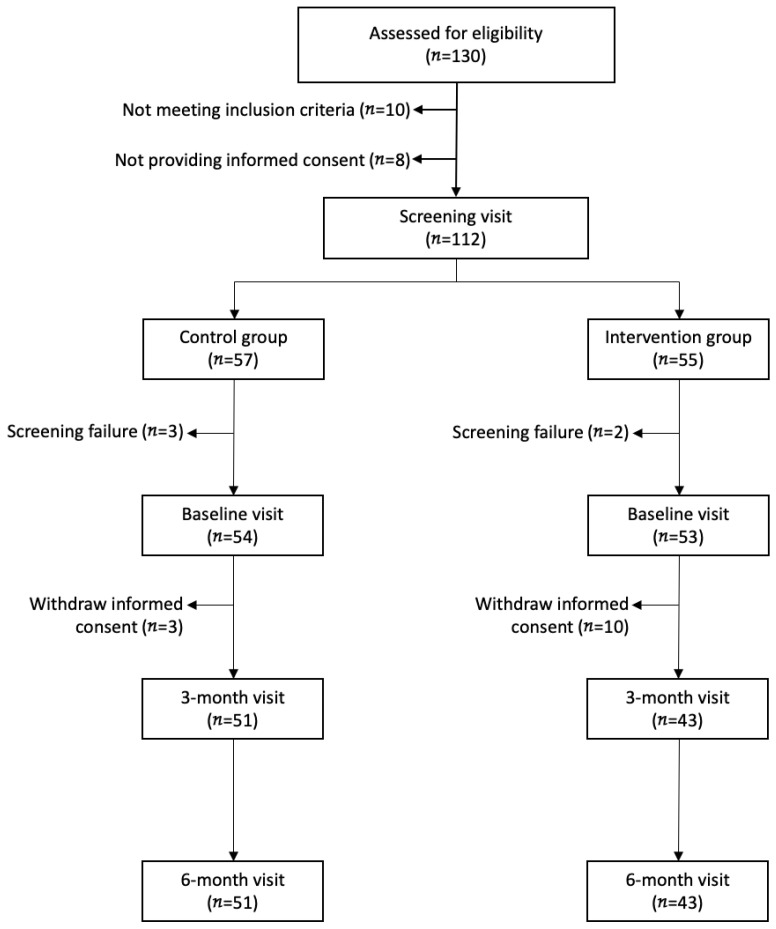
Study flowchart.

**Table 1 nutrients-17-02768-t001:** Baseline characteristics divided by study group.

Characteristic	Intervention N = 43 ^1^	Control N = 51 ^1^	*p*-Value ^2^
Age	58.4 ± 12.3 (31.4–79.2)	62.2 ± 11.4 (31.9–79.8)	0.2
Marital status			0.13
Single	7 (16%)	13 (25%)	
Partnered	27 (63%)	32 (63%)	
Separated/divorced	7 (16%)	5 (10%)	
Widowed	2 (5%)	1 (2%)	
Education level			0.4
No formal education	5 (12%)	7 (14%)	
Primary education	4 (9%)	8 (16%)	
Secondary education	11 (26%)	12 (23%)	
Vocational training	16 (37%)	19 (37%)	
University	7 (16%)	5 (10%)	
Cancer stage			0.2
I	5 (12%)	6 (12%)	
II	12 (28%)	16 (31%)	
III	22 (51%)	24 (47%)	
IV	4 (9%)	5 (10%)	
Radiotherapy			0.13
Yes	40 (93%)	40 (78%)	
No	3 (7%)	11 (22%)	
Diabetes with organ damage			0.4
No	40 (93%)	43 (84%)	
Yes	3 (7%)	8 (16%)	
Hypertension			0.08
No	11 (26%)	23 (45%)	
Yes	32 (74%)	28 (55%)	

^1^ Mean ± SD (Min–Max); *n* (%). ^2^ Welch Two Sample *t*-test; Pearson’s Chi-squared test.

**Table 2 nutrients-17-02768-t002:** Lymphedema volume and circumference changes by weight loss classification.

	Non-WL Group N = 67			WL Group N = 27		
	Baseline	3-Month Changes	6-Month Changes	Baseline	3-Month Changes	6-Month Changes
Volume (mL)						
Affected limb	5434.9 ± 1197.7	−153.55 ± 588.93	−395.58 ± 596.9	5255.37 ± 1154.8	−380.89 ± 343.63 *	−664.91 ± 362.05 #
Healthy limb	4457.55 ± 812.71	−98.58 ± 367.87	−266.38 ± 356.36	4494.44 ± 906.34	−220.96 ± 365.21	−484.91 ± 359.33 #
Limbs Difference	977.34 ± 813.57	−54.97 ± 496.11	−129.2 ± 454.45	760.93 ± 701.1	−159.93 ± 305.71	−180.0 ± 318.95
Affected arm circumferences (cm)						
Level 65%	35.06 ± 3.7	−0.27 ± 1.63	−0.96 ± 1.37	35.09 ± 3.79	−1.24 ± 1.15 **	−2.2 ± 1.2 ##
Mid Arm	34.43 ± 3.96	−0.38 ± 1.65	−1.06 ± 1.63	34.3 ± 3.87	−1.26 ± 1.16 *	−2.04 ± 1.19 #
Elbow	29.2 ± 3.34	−0.33 ± 1.65	−0.48 ± 1.83	28.57 ± 2.7	−0.93 ± 1.19	−1.43 ± 1.04 #
Forearm	27.03 ± 3.97	0.04 ± 1.76	−0.36 ± 1.72	26.3 ± 3.08	−0.70 ± 1.15 *	−1.3 ± 0.95 ##
Wrist	17.82 ± 2.67	−0.28 ± 1.98	−0.45 ± 2.85	17.37 ± 1.45	−0.54 ± 0.73 *	−0.35 ± 0.53
Healthy arm circumferences (cm)						
Level 65%	33.27 ± 3.33	−0.27 ± 1.33	−0.87 ± 1.46	33.57 ± 3.44	−1.22 ± 1.55 *	−2.09 ± 1.48 #
Mid Arm	31.96 ± 3.39	−0.46 ± 1.31	−0.98 ± 1.58	32.57 ± 3.49	−1.52 ± 1.50 **	−2.43 ± 1.77 #
Elbow	26.54 ± 2.15	−0.21 ± 1.37	−0.39 ± 1.02	26.5 ± 2.04	−0.72 ± 0.76 *	−1.04 ± 0.86 #
Forearm	23.93 ± 2.24	−0.38 ± 2.60	−0.3 ± 1.22	23.72 ± 2.14	−0.24 ± 1.25	−0.5 ± 1.25
Wrist	16.49 ± 1.17	0.08 ± 0.94	0.03 ± 0.98	16.39 ± 1.1	−0.13 ± 0.77	−0.24 ± 0.9

Data are presented as mean ± standard deviation. Results correspond to the per-protocol (PP) population, stratified according to achievement of ≥5% body weight reduction (weight loss group, WL) versus < 5% (non-WL group) (post hoc analysis). * *p* < 0.05, ** *p* < 0.001 for between-group differences in 3-month changes from baseline; # *p* < 0.05, ## *p* < 0.001 for between-group differences in 6-month changes from baseline (all from independent t-test or Mann–Whitney U test, depending on normality). Complementary intention-to-treat (ITT) analyses following the same stratification are provided in Appendix A.

**Table 3 nutrients-17-02768-t003:** Changes in morphofunctional parameters by study group.

	Baseline Values		3-Months Changes		6-Months Changes	
	Intervention N = 43	Control N = 51	Intervention N = 43	Control N = 51	Intervention N = 43	Control N = 51
Anthropometry						
BMI (kg/m^2^)	29.64 ± 2.86	31.63 ± 5.04	−1.17 ± 1.17	−0.54 ± 1.21 *	−1.14 ± 1.22	−0.53 ± 1.22 #@
Triceps Skinfold (mm)	28.54 ± 5.5	29.31 ± 6.43	−2.9 ± 2.4	−1.12 ± 1.83 **	−4.61 ± 3.02	−1.39 ± 2.44 ##@@
Waist Circumference (cm)	89.85 ± 10.58	93.37 ± 11.53	−2.93 ± 3.04	−1.16 ± 4.71 *	−3.59 ± 4.6	−2.19 ± 4.14 #@
Arm Circumference (cm)	30.05 ± 2.52	31.28 ± 3.47	0.74 ± 1.15	−0.49 ± 1.29	−1.66 ± 1.19	−0.65 ± 1.51 ##
BIA						
FFM (kg)	44.6 ± 4.82	45.44 ± 5.15	−0.69 ± 1.43	−0.67 ± 1.38	−0.76 ± 1.42	−0.23 ± 1.96 @
FFMI (kg/m^2^)	17.8 ± 1.54	18.36 ± 1.84	−0.26 ± 0.53	−0.27 ± 0.58	−0.31 ± 0.6	−0.1 ± 0.76 @
FM (kg)	27.61 ± 6.09	30.26 ± 8.9	−1.84 ± 2.32	−0.81 ± 2.24 *	−2.38 ± 2.75	−1.05 ± 2.43 #@@
ECW (l)	15.43 ± 1.53	15.94 ± 1.95	−0.41 ± 0.67	−0.26 ± 0.44	−0.58 ± 0.85	−0.24 ± 0.49 #@@
PhA (º)	4.89 ± 0.73	4.91 ± 1.05	0.02 ± 0.35	−0.05 ± 0.85	0.02 ± 0.41	−0.09 ± 0.92
MM (kg)	7.36 ± 0.69	7.71 ± 0.88	0.73 ± 5.2	−0.11 ± 0.25	−0.13 ± 0.26	−0.1 ± 0.28
FM Affected Arm (kg)	1.47 ± 0.51	1.76 ± 0.68	−0.07 ± 0.22	−0.09 ± 0.21	−0.08 ± 0.25	−0.09 ± 0.2 @
FFM Affected Arm (kg)	2.47 ± 0.54	2.59 ± 0.48	−0.02 ± 0.25	−0.03 ± 0.21	−0.1 ± 0.39	−0.06 ± 0.26
SMM Affected Arm (kg)	2.38 ± 0.48	2.49 ± 0.48	−0.04 ± 0.2	−0.05 ± 0.2	−0.09 ± 0.24	−0.05 ± 0.2
Dynamometry						
HGS (kg)	20. 5 ± 4.27	19.40 ± 5.15	1.36 ± 2.83	0.48 ± 2.28	2.37 ± 3.21	1.21 ± 3.15
QIS (kg)	18.36 ±13.39	19.39 ± 7.52	5.6 ± 6.44	−1.33 ± 4.35	7.10 ± 9.66	−3.35 ± 6.66 ##

Data are presented as mean ± standard deviation. Results correspond to the per-protocol (PP) population. Fat-free mass, FFM; Fat-free mass index, FFMI; Fat mass, FM; Extracellular water, ECW; Phase angle, PhA; Muscle mass, MM; Skeletal muscle mass, SMM; Hand-grip strength, HGS; Quadriceps isometric strength, QIS. * *p* < 0.05, ** *p* < 0.001 for between-group differences in 3-month changes from baseline; # *p* < 0.05, ## *p* < 0.001 for between-group differences in 6-month changes from baseline; @ *p* < 0.05, @@ *p* < 0.001 for between-group differences in 6-month vs. 3-month changes (all from independent t-test or Mann–Whitney U test, depending on normality). Complementary intention-to-treat (ITT) analyses are available in Appendix A.

**Table 4 nutrients-17-02768-t004:** Changes in dietary intake and adherence to the Mediterranean Diet by study group.

	Baseline Values		3-Months Changes		6-Months Changes	
	Intervention N = 43	Control N = 51	Intervention N = 43	Control N = 51	Intervention N = 43	Control N = 51
PREDIMED Score	8.57 ± 2.76	7.4 ± 3.37	1.98 ±3.55	1.8 ± 3.54 $$	0.04 ± 3.9	−1.22 ± 4.57 $$
Energy (kcal)	1874.77 ± 373.52	1845.91 ± 309.27	−290.22 ± 480.21	−43.35 ± 379.1 *	−216.22 ± 424.52	−19.03 ± 358.07 #
Proteins (g)	81.49 ± 30.69	80.43 ± 18.24	−8.73 ± 35.81	0.59 ± 27.21	16.96 ± 122.13	2.19 ± 23.43
Total Fat (g)	83.35 ± 26.01	84.39 ± 15.97	−14.88 ± 30.09	2.94 ± 22.39 *	−9.66 ± 26.18	5.32 ± 25.29 #
Total Carbohydrates (g)	192.66 ± 52.48	202.83 ± 47.31	−29.63 ± 64.31	−18.16 ± 48.89	−23.1 ± 55.38	−14.24 ± 48.49
Glycemic Load	100.28 ± 33.79	102.86 ± 28.87	−25.93 ± 37.36	−6.36 ± 35.78 *	−19.67 ± 31.46	−3.74 ± 33.37 #
Glycemic Index	57.3 ± 19.15	51.81 ± 5.96 $	−12.1 ± 21.6	0.31 ± 11.52 **	−6.87 ± 10.09	0.39 ± 7.37 ##
Fiber (g)	15.35 ± 6.39	16.46 ± 6.19	6.43 ± 6.94	1.6 ± 7.94 *	5.34 ± 6.63	3.27 ± 7.25
Saturated Fat (g)	16.73 ± 7.06	16.4 ± 4.74	−4.36 ± 7.89	1.61 ± 6.92 *	−3.74 ± 7.1	0.21 ± 6.37 #
Monounsaturated Fat (g)	33.8 ± 11.43	33.27 ± 9.66	−0.57 ± 13.31	7.33 ± 12.97 *	−0.13 ± 10.19	4.57 ± 14.43
Polyunsaturated Fat (g)	12.74 ± 7.11	11.59 ± 8.31	−2.04 ± 9.61	1.19 ± 9.58	−1.15 ± 7.7	3.09 ± 11.39
Omega 3 (g)	1.25± 1.15	1.18 ± 1.28	0.92 ± 1.46	0.2 ± 1.44 *	0.64 ± 1.39	0.37 ± 1.75
Omega 6 (g)	7.24 ± 5.96	5.11 ± 5.36	−2.15 ± 7.98	0.76 ± 8.22	−1.66 ± 6.09	2.49 ± 7.95 #
Calcium (mg)	740.52 ± 296.27	798.55 ± 294.68	−97.75 ± 326.57	−78.06 ± 339.17	−36.19 ± 245.81	−108.63 ± 276.69
Cholesterol (mg)	265.23 ± 141.29	261.74 ± 138.37	−52.85 ± 208.54	51.04 ± 210.12	−50.62 ± 175.34	28.53 ± 211.96 #
EPA (mg)	0.16 ± 0.29	0.14 ± 0.36	0.51 ± 1.45	0.07 ± 0.43	0.05 ± 0.33	0.03 ± 0.42
DHA (mg)	0.31 ± 0.67	0.27 ± 0.58	0.19 ± 0.76	0.11 ± 0.72	0.23 ± 0.71	0.07 ± 0.64

Data are presented as mean ± standard deviation. Results correspond to the per-protocol (PP) population. $ *p* < 0.05, $$ *p* < 0.001 (from repeated measures ANOVA) for within-group changes over time. * *p* < 0.05, ** *p* < 0.001 for between-group differences in 3-month changes from baseline; # *p* < 0.05, ## *p* < 0.001 for between-group differences in 6-month changes from baseline. Complementary intention-to-treat (ITT) analyses are available in Appendix A.

## Data Availability

The individual de-identified participant data that support the findings of this study are not publicly available. However, certain datasets may be made available upon reasonable request to the corresponding authors (M.P. and G.O.), subject to institutional and ethical approvals. The data available for sharing include demographic, clinical, and outcome variables collected during the trial. Additional study documents, including the study protocol and the statistical analysis plan, will also be made available upon request.

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
