# Peer review of "Targeting Lymphedema in Overweight Breast Cancer Survivors: A Pilot Randomized Controlled Trial of Diet and Exercise Intervention"

_nutrients, 2025, doi:10.3390/nu17172768_

Round 1

Reviewer 1 Report

Comments and Suggestions for Authors

The manuscript addresses an important clinical question and is based on a well-defined randomized controlled design. The multidisciplinary intervention combining a hypocaloric Mediterranean diet, supervised aerobic and resistance training, and- optional supplementation - is a strength, as is the use of robust and comprehensive outcome measurements. The post hoc analysis by ≥5% weight loss adds clinical relevance, and the observation that even basic lifestyle advice benefited the control group is noteworthy.

However, the study also has important limitations that should be clearly acknowledged. The open-label design, potential “intervention spillover” to the control group, modest effective sample size (particularly in subgroup analyses), short follow-up period, and the exploratory nature of the key weight-loss analysis limit the strength of the conclusions. Variable adherence to the intervention (especially supplementation and supervised sessions) further adds heterogeneity.

Given these factors, I strongly recommend framing this work as a pilot randomized controlled trial in both the title and the abstract. For example: Targeting lymphedema in overweight breast cancer survivors: A pilot randomized controlled trial of diet and exercise intervention. I also recommend adding a dedicated paragraph in the Limitations section explicitly stating that the sample size, short follow-up, and post hoc analyses make this a preliminary study, with results that require confirmation in larger, longer-term trials.

Overall, the manuscript is well written, the English is clear, and the results will be of interest to clinicians and researchers in the field. The suggested adjustments will help ensure that readers interpret the findings appropriately.

Reviewer 2 Report

Comments and Suggestions for Authors

This manuscript describes the results of a well-designed randomized controlled trial (RCT) evaluating the effects of a comprehensive lifestyle intervention on breast cancer-related lymphedema (BCRL) in overweight women. The authors conclude that a ≥5% weight loss is associated with significant reductions in limb volume and improvements in morphofunctional outcomes, but not systemic inflammatory markers.

The study addresses a highly relevant issue. The overall design is robust, and the outcome measures are relevant. However, there are a few points that need to be addressed.

There does not seem to be a distinction between intention-to-treat (ITT) analysis (considered to be the method least prone to bias) and per-protocol analysis (the method used in this manuscript, apparently). Results from both methods of analysis should be presented.

Furthermore, the main emphasis of the analysis seems to be based on those participants who achieved overall weight loss versus those who did not, regardless of randomised intervention. These post-hoc analyses should be labelled as such and presented after the ITT and per-protocol analyses. 

It is concerning that the information in the clinicaltrials.gov entry (https://clinicaltrials.gov/study/NCT04974268) does not seem to fully correspond with the analyses described in the manuscript. In addition, the protocol has not been made available.

Adherence to the meal replacement was measured by self-reported questionnaire and has been reported, but not the other components of the intervention. 

The authors have clearly gone to a great deal of effort to run this RCT, but a more comprehensive analysis of the results needs to be presented.

Round 2

Reviewer 2 Report

Comments and Suggestions for Authors

The comments and suggestions made previously have been addressed.